# Lifestyle and Environmental Factors Affecting Male Fertility, Individual Predisposition, Prevention, and Intervention

**DOI:** 10.3390/ijms26062797

**Published:** 2025-03-20

**Authors:** Jan Tesarik

**Affiliations:** MARGen (Molecular Assisted Reproduction and Genetics) Clinic, Calle Gracia 36, 18002 Granada, Spain; jantesarik13@gmail.com; Tel.: +34-606376992

**Keywords:** male fertility, smoking, alcohol, stress, physical activity, obesity, electronic devices, air pollution, heat, harmful chemicals, genetics, epigenetics, systemic disease, infection, prevention, treatment

## Abstract

Current lifestyles bring about an increasing prevalence of unhealthy habits that can negatively affect male fertility. Cigarette smoking, alcohol intake, stress, inadequate physical activity, an unequilibrated diet leading to obesity, and use of mobile telephones and portable electronic devices can affect the male reproductive system through multiple mechanisms. Moreover, the modern man is often exposed to environmental factors independent of his will, such as air pollution, exposure to heat or toxicants in his workplace, or the presence of harmful chemicals in food, beverages, agricultural and industrial products, etc. The susceptibility to these factors depends on genetic and epigenetic predisposition, potentially present systemic disease and medication, and local affections of the genitourinary system. The multifaceted nature of both the causative factors and the susceptibility background makes the resulting fertility disturbance highly individual and variable among different men exposed to the same conditions. This paper critically reviews the current knowledge of different causative and susceptibility factors with a special attention to the molecular mechanisms of their action. Finally, strategies for the prevention of abnormalities due to lifestyle and environmental factors and available treatment modalities for already-present abnormalities are exposed.

## 1. Introduction

Lifestyle factors, also referred to as behavioral factors, are those related to the change in people’s behaviors and the way they live their life. Together with environmental factors, they can be at the origin of a wide range of noncommunicable diseases (NCDs), such as cardiovascular diseases, type 2 diabetes, chronic obstructive pulmonary disease, and some types of cancer, or the risk factors behind them [1]. The prevalence and incidence of NCDs shows an increasing trend in different parts of the world [2,3,4,5]. In addition to the above systemic NCDs, lifestyle and environmental factors were also reported to affect male fertility [6,7]. Environmental factors implicated in the pathogenesis of male subfertility and infertility act on the background of each individual’s susceptibility, which is defined by genetic predisposition, the occasional presence of a systemic disease, local affections of the genitourinary system, and others. This review deals with the most important lifestyle and susceptibility factors, their molecular mechanisms of action, and the current possibilities of prevention and intervention.

The choice of the studies included in this review article was based exclusively on their novelty and scientific impact. Both meta-analyses and original research studies were valued equally according to these criteria. Eventual conflicting results were mentioned and discussed.

## 2. Contemporary Lifestyle and Male Fertility

There has been a continuous decline in semen quality, including a decrease in semen volume, sperm count, concentration, and the percentage of normal forms, over the last decades [8,9,10]. This trend is supposed to be due to the negative influence of widespread lifestyle-related habits to which the modern man is exposed during his reproductive period [6]. The main negative lifestyle habits and environmental conditions affecting male fertility include an inadequate diet leading to obesity, air pollution, exposure to harmful chemicals (food and drinks, agricultural and industrial products), exposure to heat, cigarette smoking, alcohol intake, stress, inadequate physical activity, and use of mobile telephones and portable computers [6,7,11]. Apart from possible effects on erectile function, most of the lifestyle and environmental factors affecting male fertility impact different aspects of sperm structure and function.

Seminal oxidative stress [12,13,14] is the main endpoint to which the molecular effects of different harmful factors converge. Oxidative stress arises from conditions in which the production of reactive oxygen species (ROS), needed for normal sperm development and function, exceeds the ROS-scavenging capacity of inherent antioxidative systems. The resulting excessive ROS produce a chain of events leading to the damage of sperm lipids, proteins, and DNA, ultimately affecting male fertility [15,16]. Basically, a spermatozoon consists of a sperm head and sperm tail. The head contains a sperm nucleus with its DNA and is covered by the acrosome. The main functional parts of the sperm tail are the midpiece and the principal piece. Mitochondria, contained in the midpiece, produce energy in the form of adenosine triphosphate (ATP), which is mainly used to sustain propulsive forces for sperm cell movement that are generated in the axoneme of the principal piece. In addition, ATP is used to power other energy-consuming sperm functions, such as the exocytosis of sperm acrosome (acrosome reaction) required for sperm penetration into the oocyte at fertilization. Harmful external factors can act directly on the sperm acrosome and nuclear DNA. They also can disturb mitochondrial function, leading to oxidative stress (see above), which, in its turn, feeds back into mitochondrial integrity and aggravates nuclear and acrosomal damage (Figure 1).

### 2.1. Obesity

Obesity and overweight prevalence is increasing globally and contributes to various human health problems including male infertility [11,17]. Obesity was reported to impair both conventional and biofunctional sperm parameters, and to induce epigenetic changes that can be transferred to offspring [11,17,18]. There are several potential biological mechanisms linking central obesity to reduced semen quality. They include a decrease in serum sex hormone-binding globulin (SHBG), total and free testosterone (T), and inhibin B levels, and increased conversion of T into 17β-estradiol (E2) [19]. Reduced hepatic insulin clearance, due to increased delivery of free fatty acids released from the adipose tissue to the liver, leads to hyperinsulinemia and insulin resistance [18]. Hyperinsulinemia dysregulates the endocrine activity of hypertrophied adipocytes by stimulating the production of non-esterified fatty acids, which increase the secretion of tumor necrosis factor α (TNFα) by the adipocytes. TNFα, in turn, potentiates the secretion of adipokines (mainly leptin) and other pro-inflammatory cytokines. The pro-inflammatory cytokines, mainly interleukin 1β (IL1β) and interleukin 6 (IL6), as well as acute-phase proteins and chemokines, mainly C-C motif chemokine ligand-2 (CCL2) and monocyte chemoattractant protein-1 (MCP-1), attract more monocytes/macrophages within the adipose tissue [20]. The resulting vicious cycle promotes the inflammation of the adipose tissue first and then a systematic state of low-grade inflammation [21]. The increased production of adipokines and pro-inflammatory cytokines significantly influences insulin signaling in insulin-responsive tissues, leading to systemic insulin resistance and consequent disturbance of gonadotropin-releasing hormone (GnRH) secretion by hypothalamic neurons [22]. In addition to acting through endocrine imbalance, insulin resistance also inhibits spermatogenesis by increasing nuclear and mitochondrial DNA damage [23]. A pivotal role is also played by heat-induced damage due to fat accumulation in the suprapubic region and around the pampiniform plexus. The resulting increase in scrotal temperature reduces sperm concentration and motility and enhances oxidative stress and sperm DNA fragmentation [24].

As to the epigenetic changes that can be transferred to offspring, epidemiological studies have shown that children born from obese fathers are more likely to be obese [25]. Among the most relevant epigenetic mechanisms that are involved in gene activity regulation (DNA methylation, histone modifications, and non-coding RNAs), DNA methylation was the most widely studied in the context of obesity [18,26]. Significantly altered DNA methylation in the regulatory region of many imprinted genes has been reported in spermatozoa from overweight and obese men as compared to normal-weight men [27]. The main genes that undergo abnormal epigenetic modification (hypomethylation) of differentially methylated regions in obese males include maternally expressed gene 3 (*MEG3*), necdin (*NDN*), small nuclear ribonucleoprotein polypeptide N (*SNRPN*), and sarcoglycan epsilon (*SGCE*)/paternally expressed gene 10 (*PEG10*), while a slight increase in DNA methylation has been detected in the *H19* gene [28]. The hypomethylation of imprinted genes was associated with higher levels of seminal oxidative stress, sperm DNA fragmentation, and decreased pregnancy rates [29,30].

### 2.2. Air Pollution

Air pollution, mainly arising from motor vehicle exhaust, factories, fire, households, agriculture, waste treatment, oil refineries, and natural sources, such as volcanic eruptions, is increasing steadily [31]. In addition to causing a number of unrelated diseases, it also affects male fertility [32,33,34]. The main air pollutants include particulate matter, volatile organic compounds, ozone, nitrogen oxides, sulfur dioxide, carbon monoxide, polycyclic aromatic hydrocarbons (PAHs), and various types of radiation (such as X-ray exposure) [35]. The impairment of several parameters of male reproductive function in response to various air pollutants has been reported. In 2018, a critical review of 22 studies dealing with the effects of various air pollutants on semen quality suggested that all of the pollutants studied (airborne particular matter, nitric oxides, sulfur dioxide, PAHs, ozone, carbon monoxide, heavy metals) can affect sperm motility, morphology, and DNA integrity [36]. Exposure to air pollutants is often occupational, for instance in men working as toll collectors at motorways. One study has reported a significant decrease in total sperm count, total and progressive motility, and the percentage of spermatozoa with a normal morphology, normal chromatin, and intact DNA in toll collectors as compared to unexposed healthy men [37]. A significant direct correlation was found between spermatozoa with damaged chromatin or fragmented DNA and the length of occupational exposure, suggesting a time-dependent relationship [37].

The mechanisms by which air pollutants affect male fertility are not completely clear and have only been studied for some of the pollutants (PAHs, heavy metals, particular matter, sulfur dioxide, carbon monoxide, nitric oxides). PAHs and heavy metals (lead, zinc, copper) from car exhaust can have estrogenic, antiestrogenic, and androgenic actions, and they may also cause hormonal disruption, leading to abnormal gonadal steroidogenesis and spermatogenesis [38,39]. Furthermore, exposure to PAHs can also result in changes in gene expression and DNA methylation [40]. Particular matter, sulfur dioxide, carbon monoxide, and nitric oxides stimulate the production of reactive oxygen species, which cause oxidative stress, leading to lipid peroxidation and sperm DNA fragmentation [39]. Further research into the molecular mechanisms underlying the effects of air pollutants on male fertility is warranted.

### 2.3. Harmful Chemicals (Other than Air Pollutants)

Food, drink, and various agricultural and industrial products are the main sources of potentially harmful chemicals to which the contemporary man can be exposed. The main harmful chemicals and their sources have been reviewed recently [31]. They include dioxins, bisphenols, pesticides and herbicides, phthalates, and heavy metals (Table 1).

Dioxins are highly persistent by-products of several industrial processes, such as smelting, chlorine bleaching of paper, production of some pesticides, or biomedical and plastic waste incineration [35]. Their effects were best studied in residents of Seveso (Italy) exposed to 2,3,7,8-tetrachlorodibenzo-*p*-dioxin released accidentally from a trichlorophenol manufacturing plant in 1976 [41]. Negative effects on sperm quality (count, motility, progressive motility) were observed mainly after exposure of children. Dioxins act as endocrine disruptors, and their toxic effects are mediated by binding to the aryl hydrocarbon receptor (AHR)/aryl hydrocarbon receptor nuclear translocator (ARNT) receptor complex present in human testicular cells [42].

Bisphenols are a group of industrial chemical compounds related to diphenylmethane, commonly used in the creation of plastics and epoxy resins; they are released into the environment during the process of production, use, or disposal of plastics and result from the breakdown of plastic-related wastes [35]. In particular, bisphenol A (BPA) was shown to affect male fertility as an endocrine disruptor. It impairs spermatogenesis by antagonizing the adrogen receptor, and it also displays estrogenic and antithyroid actions [43,44]. Other studies reported BPA disturbing sperm function through inducing a premature sperm acrosome reaction and decreasing sperm motility by reducing adenosine triphosphate (ATP) levels in spermatozoa [45]. Moreover, BPA causes activation of pro-apoptotic signaling pathways, including mitogen-activated protein kinase (MAPK), Fas/FasL, Caspases 3 and 9, and Bax, leading to diminished proliferation, increased reactive oxygen species-mediated damage, and enhanced apoptosis of male germ cells [46]. Exposure to BPA was also shown to lead to sperm DNA damage, mitochondrial dysfunction and degeneration, and increased risk of aneuploidies in spermatozoa [44,47].

Pesticides can cause direct sperm damage or disrupt the endocrine function needed for spermatogenesis at any stage of hormonal regulation (hormone synthesis, release, storage, transport and clearance, receptor recognition and binding); in addition to the reproductive system, they can also affect thyroid function and the central nervous system [48]. The endocrine-disrupting action of organochlorine pesticides, the most widely used ones, is well-documented [49]. The use of dichlorodiphenyltrichloroethane (DDT), the most notorious organochlorine pesticide, is currently restricted in most countries. However, due to its stability and capacity to accumulate in the adipose tissue, DDT can persist in the human organism even for decades after exposure, leading to the conclusion that “there is now not a single living organism on the planet that does not contain DDT” [49]. Most of the endocrine-disrupting action of DDT is due to its main metabolite, *p*,*p*′-dichlorodiphenyl-dichloroethylene (*p*,*p*′-DDE), which binds to androgen receptors and thus inhibits the action of testosterone [50]. *p*,*p*′-DDE may have an additive or multiplicative effect with other endocrine-disrupting environmental pollutants, potentiating their adverse impacts on reproductive functions [49]. Exposure of spermatozoa to *p*,*p*′-DDE was also reported to promote mitochondrial Ca^2+^ overload, a decrease in mitochondrial membrane potential, an increase in reactive oxygen species (ROS) production, and sperm DNA fragmentation, which collectively lead to a general mitochondrial dysfunction and cellular ATP depletion with a remarkable decrease in sperm motility and fertilizing ability [51,52]. DDT and other organochlorine chemicals enter the food chain and can be transported over long distances [53].

**Table 1 ijms-26-02797-t001:** Overview of the most important harmful chemicals, their sources, effects, and the mechanisms of action. Up/down arrows stand for upregulating and downregulating effects, respectively.

Name	Sources	Effects	Mechanism of Action	References
**Dioxins**	Industrial accidents	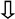 Sperm count 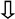 Sperm motility	Endocrine disruption	[41,42]
**Bisphenols**	Plastics	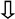 Spermatogenesis 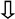 Sperm motility 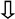 Sperm function 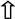 Sperm apoptosis 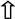 Sperm aneuploidy	Endocrine disruptionOxidative stress 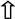 MAPK 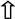 Fas/FasL 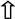 Caspases 3, 9 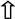 Bax 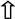 DNA damage 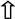 Mitochondrial damage	[44,45,46,47]
**Pesticides and herbicides**	Agricultural products	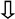 Spermatogenesis 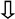 Sperm motility 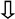 Sperm count 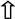 Sperm apoptosis	Endocrine disruptionOxidative stress 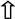 DNA damage 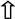 Mitochondrial dysfunction	[48,49,50,51,52,53,54,55]
**Phthalates**	Variousindustrial goods	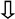 Spermatogenesis 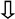 Sperm motility 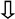 Sperm count 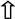 Teratozoospermia	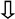 Endocrine disruption 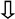 Gene transcription Androgene synthesis	[56,57,58,59,60,61,62,63,64]
**Heavy metals**	Variousindustrial goods	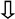 Spermatogenesis 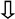 Sperm motility 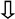 Sperm count 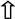 Teratozoospermia 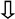 Sperm viability	Endocrine disruptionOxidative stress 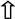 DNA damage 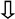 Androgene synthesis	[65,66]

Herbicides have received less attention than pesticides in academic publications. In fact, published data only concern one herbicide named Roundup. Direct exposure of human spermatozoa to the active component of this herbicide (glyphosate), acting as an endocrine disrupter, was reported to cause sperm mitochondrial dysfunction and a decrease in sperm motility [54]. Another study, carried out in the rat model, analyzed the molecular mechanisms of glyphosate action in the testis and isolated Sertoli cells [55]. It was shown that glycosate increases intracellular Ca^2+^ concentration by opening L-type voltage-dependent Ca^2+^ channels as well as endoplasmic reticulum inositol trisphospate (IP3)- and ryanodine receptor-operated channels, leading to Ca^2+^ overload within the rat prepubertal Sertoli cells, which sets off oxidative stress and necrotic cell death. These effects were prevented by the antioxidants Trolox and ascorbic acid [55]. Activated protein kinase C, phosphatidylinositol 3-kinase, and the mitogen-activated protein kinases, such as ERK1/2 and p38MAPK, were shown to play a role in eliciting the glycosate-induced Ca^2+^ influx. Also, glyphosate decreases the levels of reduced glutathione (GSH) and increases the amounts of thiobarbituric acid-reactive species (TBARS) and protein carbonyls. In fact, simultaneous stimulating effects on the activities of glutathione peroxidase, glutathione reductase, glutathione S-transferase, γ-glutamyltransferase, catalase, superoxide dismutase, and glucose-6-phosphate dehydrogenase potentiate the consequences of downregulated GSH levels, further decrease GSH, and increase the amounts of TBARS and protein carbonyls [55].

Phthalates, also known as phthalic acid diesters, are man-made chemicals that are used in several consumer and industrial goods [56]. They were found in many consumer products such as toys, pharmaceuticals, cosmetic products, building and construction materials, scent retainers, some medications, personal care products, etc. [57]. Male partners of infertile couples, who have been exposed to phthalate drugs, have poor sperm quality as compared to unexposed men [57,58]. In particular, exposure to phthalates decreases semen volume, total sperm count, and sperm concentration, and causes diverse morphological abnormalities of the sperm head [59,60], in addition to reducing sperm motility [60,61]. At the molecular level, phthalates, especially mono-(2-ethyl-hexyl) phthalate (MEHP), an active metabolite of di-2-ethylhexyl phthalate (DEHP), cause activation of peroxisome proliferator-activated receptor (PPAR) α and γ pathways [62]. The activation of both pathways stimulates PPAR:RXR (retinoid X receptor) heterodimers that compete for the DNA binding sites required for gene transcription, impeding the synthesis of the aromatase enzyme involved in sexual development [63]. In addition, MEHP decreases the production of steroidogenic proteins including steroidogenic acute regulatory (StAR) and cytochrome P450 side-chain cleavage (P450scc) proteins [31]. At high levels, it also inhibits the activity of 3β-hydroxysteroid dehydrogenases (3β-HSD) and 17β-hydroxysteroid dehydrogenases (17β-HSD) specific to Leydig cell function by increasing oxidative stress in Leydig cells; this inhibitory action leads to a decrease in testosterone synthesis [64].

Heavy metals, such as lead, cadmium, arsenic, mercury, barium, uranium, and others, were also reported to reduce the semen and sperm quality in men. Data were published showing a close association between the presence of cadmium and barium in blood, and lead, cadmium, barium, and uranium in seminal plasma, on the one hand, and increased risk for a reduced sperm count, viability, and motility, and abnormal sperm morphology, on the other hand [65]. An increase in reactive oxygen species (ROS) generation, leading to excessive lipid peroxidation and sperm DNA damage, is the main mechanism by which heavy metals affect male infertility [64], although they also act as endocrine disrupters, which affect semen quality by altering the production of androgenes and their action in the testis [31].

### 2.4. Heat

Excessive heat exposures in the workplace, or those caused by elective lifestyle habits, are known factors affecting spermatogenesis. The question of whether global warming also can play a role is to be evaluated with interest in future studies. In fact, any factor that causes a rise in scrotal temperature (normally 2–4 °C lower than the core body temperature) can be blamed [67,68,69]. Even moderate scrotal temperature rise (1–1.5 °C) was shown to impair sperm production (oligozoospermia, azoospermia) and to cause sperm morphological abnormalities (teratozoospermia) [70]. Alteration in spermatogenesis in men with a varicocele is actually related to scrotal temperature increase [68].

Exposure to heat can be self-imposed or nonelective, professional or environmental, transient or durable. Self-imposed exposures are related to elective lifestyle habits. As a matter of example, a study has revealed that the frequent use of tight underwear in men can increase scrotal temperature, leading to a decreased sperm concentration, total sperm count, and motility, and male infertility [71]. Frequent exposure to wet heat (hot tubs, Jacuzzis, etc.) also affects sperm count and motility and can disturb fertility in males [72,73]. Unlike self-imposed heat exposure, there are also conditions of nonelective exposures, mainly professional and environmental ones. Professional exposure to radiant heat, as seen in cases of people working in furnaces, bakeries, and welding or ceramic factories, and in those working for long hours in kitchens, laundries, or dry cleaning, was shown to affect one or more components of semen quality in males [69,70,74]. Chronic environmental exposure to heat has been increasing over the past decades due to climate change. An inverse relationship between ambient temperature and sperm quality has been documented. Sperm concentration, motility, and total amount per ejaculate are significantly lower in summer and higher in winter [75]. Another study, also including sperm motility evaluation, reported similar trends and suggested the winter and spring semen patterns to be compatible with increased fecundability, which may be a plausible explanation of the peak number of deliveries during the fall [76]. The consequences of self-imposed heat exposure are usually reversible and disappear when the causative lifestyle habit is corrected. This was demonstrated in some, but not all, patients who responded favorably to cessation of voluntary heat exposure (hot baths, Jacuzzis) [72,73]. On the other hand, it is evident that nonelective exposures, both professional ones and those related to non-modifiable environmental factors, are more difficult to control.

There appear to be several mechanisms mediating the effect of heat on spermatozoa. It was observed that higher scrotal temperatures result in a rise in testicular metabolism without the corresponding surge in blood supply, leading to local tissue hypoxia and oxidative stress [77]. Owing to the high levels of polyunsaturated fatty acids in the sperm plasma membrane [78], exposure to oxidative stress causes increased production of reactive oxygen species, leading to sperm DNA fragmentation [79]. Independently, excessive heat exposure decreases sperm motility by downregulating mitochondrial activity via activation of glycogen synthase kinase-3α and inhibition of mitochondrial protein import, thus reducing ATP levels [80]. Finally, even a transient rise in scrotal temperature decreases the level of the anti-apoptotic *B cell lymphoma 2* (*Bcl-2*) gene expression, which facilitates sperm apoptotic pathways [81].

### 2.5. Cigarette Smoking

Tobacco smoke is a toxic and carcinogenic mixture of more than 5000 chemicals [82], some of which, including nicotine and its metabolites, carbon monoxide, benzopyrene, and cadmium, may have harmful effects on male germ cells [6]. A meta-analysis comprising 5865 men shows that cigarette smoking is associated with reduced sperm count and motility as well as overall deterioration in semen quality, this trend being more pronounced in moderate and heavy smokers than in occasional ones [83]. In addition to all conventional semen parameters, sperm chromatin condensation and sperm viability are also affected in smokers proportionally to the number of cigarettes smoked per day and the duration of smoking [84,85]. Evaluation of the mechanisms underlying the effects of smoking on spermatozoa is complicated because of the presence of multiple potentially harmful chemicals in cigarette smoke. Among them, polycyclic aromatic hydrocarbons (PAHs) display a preeminent role in accelerating germ cell death across all stages of spermatogenesis. This effect is mediated by the cytoplasmic transcription factor aryl hydrocarbon receptor (AHR), whose interaction with cigarette smoke components adversely affects the expression of genes associated with antioxidant mechanisms, cell proliferation and apoptosis, and cell cycle progression [86]. Additionally, cigarette smoke-mediated crosstalk between AHR and nuclear factor erythroid 2-related factor 2 (NRF2), a transcription factor that is implicated in activating the expression of genes involved in the response to different cellular insults, and between AHR-NRF2 and mitogen-activated protein kinase (MAPK) pathways induces cell death of spermatocytes [86]. Animal studies (rat) suggest that, like traditional smoking, chemicals derived from electronic cigarette vapors also have adverse effects on male fertility and semen parameters [87].

### 2.6. Alcohol Intake

Chronic heavy drinking, especially when associated with cigarette smoking, is linked to lower testosterone levels, leading to impaired semen quality, while moderate alcohol consumption may not have significant adverse effects [88]. Daily use of alcohol by men of reproductive age was shown to affect semen volume and sperm morphology [89], to cause hormone imbalance [90], and to lengthen the time to pregnancy [91]. The mechanisms of the damaging impact of alcohol on fertility are not yet fully discovered. Alcohol drinking was reported to cause a hormonal shift towards a higher free estradiol/free testosterone ratio [92] and to augment the level of reactive oxygen species and DNA damage in male germ cells [93].

### 2.7. Psychological Stress

Stress can result from both physical and psychological conditions. Physical stress is dealt with in Section 2.8 of this article. Prolonged exposure to psychological stress can reduce testosterone levels, impair sperm production, and decrease libido, in addition to promoting the acquisition of unhealthy behaviors, such as poor diet, lack of exercise, smoking, and excessive alcohol consumption [94]. Elevated levels of stress hormones, particularly cortisol, were shown to interfere with the secretion of gonadotropin-releasing hormone (GnRH) from the hypothalamus, thus decreasing the release from the pituitary gland of luteinizing hormone (LH) and follicle-stimulating hormone (FSH), both essential for stimulating the testis to produce the testosterone needed for spermatogenesis [95,96]. High cortisol levels can also make germ cells more vulnerable to oxidative stress, causing sperm DNA damage and a reduction in sperm motility and viability [97].

### 2.8. Inadequate Physical Activity

Research suggests that both insufficient and excessive physical activity can cause male fertility issues, although the connection with male fertility is complex and multifaceted [98]. Resistance exercise training was reported to improve markers of male fertility and reproductive capacity in infertile patients [99], while excessive exercise, such as intense training for marathons or Olympic events, can potentially lower testosterone levels, reduce sperm quality, and lead to infertility [100]. At appropriate levels, physical exercise attenuates inflammation, as reflected by reduced seminal pro-inflammatory cytokines (IL1β, IL6, IL8, and TNFα), decreases the levels of oxidative stress markers (malondialdehyde and 8-isoprostane), and enhances the production of antioxidants (superoxide dismutase and catalase) that protect sperm DNA integrity [99]. However, the available studies have limitations, and more research is needed to fully understand the specific effects of different regimens of exercise and their impact on fertility status [98].

### 2.9. Use of Mobile Telephones and Portable Computers

Mobile telephones and other similar electronic devices may affect male fertility both through a thermal effect and a nonthermal effect. The thermal effect is related to the fact that mobile phones are often carried in trouser pockets near external male reproductive organs [101], and portable computers are often placed on the user’s lap (laptop computers). The consequences of testis exposure to heat are reviewed in Section 2.4 of this article. The nonthermal effects of portable electronic devices are due to low-level radio-frequency electromagnetic fields (RF-EMF) [6]. A meta-analysis of ten studies, including 1492 samples, has concluded that exposure to RF-EMF negatively affects parameters of human sperm quality, mainly sperm motility and viability, while effects on sperm concentration were more equivocal [102]. Another meta-analysis, including eighteen studies with 3947 men and 186 rats, indicated that RF-EMF had detrimental effects on sperm motility and viability in human in vitro studies and on sperm concentration and motility in animal studies, while no relationship with semen parameters was found in human in vivo studies [103]. It was also reported that ex vivo exposure of human spermatozoa to a wireless internet-connected laptop decreased motility and induced DNA fragmentation by a nonthermal effect [104,105]. It was suggested that sperm exposure to electromagnetic fields may cause oxidation of phospholipids and produce high seminal reactive oxygen species (ROS) levels [6,104], but further research is still needed.

## 3. Individual Susceptibility to Negative Lifestyle and Environmental Factors

Susceptibility to negative lifestyle and environmental factors is highly individual and mainly conditioned by genetic predisposition, sperm epigenetics, systemic disease, and local affection of the genitourinary system (Table 2).

### 3.1. Genetic Predisposition

Genome alterations can cause different pathological phenotypes. However, the penetrance of a number of genome alterations is incomplete and the development of the corresponding phenotype depends on other (epigenetic and environmental) factors. In other words, genetic and epigenetic predisposition can help explain why certain individuals exposed to the same load of a lifestyle or environmental harmful factor exhibit symptoms or signs of disease, while others do not [106]. For instance, men who are homozygous null for the glutathione S-transferase mu 1 (*GSTM1*) gene have a lower capacity to detoxify reactive metabolites of polycyclic aromatic hydrocarbons (PAHs) and are consequently more susceptible to the effects of air pollution on sperm chromatin [107]. Research shows that men with *GSTM1*-null and glutathione S-transferase theta 1 (*GSTT1*)-present genotypes are susceptible to infertility, particularly that resulting from 4-*n*-octylphenol exposure [106]. Moreover, polymorphism in several DNA repair genes, such as X-ray repair cross-complementing protein 1 (*XRCC1*), xeroderma pigmentosum group D 6 (*XPD6*) and 23 (*XPD23*), and cytochrome P450 1A1 (*CYP1A1*), was observed to be associated with high or medium DNA damage after exposure to air pollution [107]. Thus, understanding the underlying genetic mechanisms related to the pathophysiology of male infertility and the impact of environmental exposures and lifestyle factors on gene expression might aid clinicians in developing individualized treatment strategies [108].

Up to 10% of cases of male subfertility and infertility were shown to have a genetic background [109]. The karyotype can identify numerical chromosome abnormalities, such as Klinefelter’s syndrome (XXY), and other male infertility-associated anomalies, such as reciprocal and Robertsonian translocations [110]. Additional molecular analyses are required, even routinely, to identify variants of genes in both the Y chromosome and the autosomes of infertile men. A recent article reviews the known genes in which mutations or gene expression changes are associated with different infertility phenotypes, including asthenozoospermia, multiple morphological anomalies of the sperm flagellum, nonobstructive azoospermia, obstructive azoospermia, oligozoospermia, and teratozoospermia [111]. According to the phenotype observed in each individual patient, the screening for deletions in these genes is highly recommended. Sperm mitochondrial DNA (mtDNA) is also prone to mutations and deletions, most of which are correlated with elevated oxidative stress and sperm immotility [112,113,114]. The content of mtDNA in spermatozoa, measured as the ratio between the amount of mtDNA and nuclear DNA, may be used as an indicator for predicting the implication of sperm mitochondria in male infertility [115].

### 3.2. Epigenetic Factors

Various epigenetic factors regulate genes in male germ cells, and alterations in sperm epigenetics, potentially influenced by environmental exposures, can contribute to male infertility [110,116] and congenital disorders in progeny [110,116,117,118,119,120]. Epigenetic markers in mature spermatozoa include post-translational histone modifications, protamines, small non-coding RNA, and DNA methylation, and can be considered as a network that aims to establish and maintain genes’ expression status [110,116,118]. A list of imprinted genes whose methylation status (hypomethylation or hypermethylation) is associated with different pathological sperm conditions has been published recently [110]. Among the imprinted genes, *H19* has been highlighted as a potential biomarker for developmental defects in human spermatozoa [121]. Actually, 100% methylation of the *H19* imprinting control area was shown in normal individuals, while *H19* was hypomethylated in diabetic infertile males [122]. Furthermore, a significant positive correlation was observed between the degree of *H19* hypomethylation and decrease in sperm count and motility, and the risk was potentiated by smoking [123]. Other environmental factors, such as endocrine-disrupting chemicals (EDCs), and lifestyle elements also have the potential to affect germline epigenetic markers, with sperm cells being the ultimate recipients of these changes and potential carriers of induced epimutations across generations through a mechanism known as paternal transgenerational epigenetic inheritance [124].

### 3.3. Systemic Disease and Medication

Interactions between systemic disease and lifestyle factors that influence semen quality have been mainly studied in the context of insulin resistance and diabetes, autoimmunity, systemic infectious diseases, and diseases requiring chronic medication.

Insulin resistance (IR) is a condition in which insulin circulating in blood is unable to properly stimulate glucose uptake and/or use by insulin-sensitive organs and tissues, leading to a compensatory increase in production of insulin in the pancreas and glucose in the liver [125]. IR is known to potentiate the adverse effects of many lifestyle and environmental factors on male fertility [126]. Indexes calculated on the basis of fasting insulin and glucose levels, namely the homeostatic model assessment of insulin resistance (HOMA) index and Matsuda index, are the best and most extensively validated indicators for IR diagnosis [127]. Both isolated and accompanied by the action of external factors, IR reduces the production of testosterone, semen volume, and the percentage of progressive spermatozoa [128]. Inversely, many lifestyle and environmental factors, such as obesity, smoking, lack of sleep, or excessive physical activity, influence both IR and male fertility [16,129,130,131].

Mechanisms of effects on male fertility (marginally mentioned in Section 2.1 of this article) are similar for IR and diabetes and have been reviewed recently [132,133]. Briefly, hyperglycemia reduces the sensitivity of the pituitary gland to stimulation by the hypothalamic gonadotropin-releasing hormone (GnRH), which reduces the secretion of FSH and LH by the pituitary gland and, consequently, disturbs spermatogenesis and testosterone production by the testis [133]. Also, the state of high blood glucose can shift the balance between oxidation and antioxidant defense in the body in favor of oxidation (oxidative stress) [134], resulting in inflammatory infiltration of neutrophils, increased protease secretion, and the production of a large amount of reactive oxygen species (ROS) and reactive nitrogen species (RNS) [133]. The hyperproduction of ROS and RNS causes oxidative damage in nuclear and mitochondrial DNA and activates the inflammatory signaling cascade in endothelial cells, releasing a large number of pro-inflammatory cytokines into blood [135,136]. Diabetes also affects epigenetic modifications in spermatogenesis [133] and may dysregulate imprinted genes in male germ cells (see Section 3.2 of this paper). In addition to reducing semen quality, diabetes can also cause erectile dysfunction [137,138].

Autoimmunity is another known systemic condition affecting spermatozoa. Antisperm antibodies, found in roughly 6% of infertile men [139], can significantly diminish sperm motility through agglutinating spermatozoa at specific regions (head to head, head to midpiece, head to tail, or non-specific binding), due to the binding of immunoglobulins to sperm surfaces [140]. This autoimmune condition can be idiopathic but is mostly found in homosexual men and patients with a varicocele, testicular trauma, mumps, orchitis, congenital absence of the vas deferens, and spinal cord injury, as well as in those who have undergone a vasectomy [141].

As to systemic infectious diseases, several bacterial and viral infections are associated with semen quality issues. Among sexually transmitted pathogens that commonly impair *parameters of sperm quality, Human papiloma virus* (*HPV*) [142,143], *Herpes simplex virus* (*HSV*) [144], *Ureaplasma urealyticum* [145], *Chlamydia trachomatis* [146], *Coronavirus* disease 2019 (COVID-19) [147,148], *Zika virus* [149], and *Adeno-associated virus* (*AAV*) [150] have been pointed out. It has been suggested that microorganisms usually affect the normal function of the male reproductive system through immune responses [151]. In many cases, the numbers of immune cells, and especially mast cells, are increased, and the proliferation of mast cells in the testis interstitium and in the walls and lumens of seminiferous tubules, along with collagen fiber deposition, hinders sperm production [152]. Bacterial infection also increases CD3 helper lymphocytes, B cells, and natural killer (NK) cells (CD56), leading to an increased level of antisperm antibodies and NK cell-mediated sperm damage [153].

Chronic medication, used to control some systemic diseases, can also be associated with male infertility [154]. In particular, there is sufficient evidence indicating a deleterious effect of chemotherapeutic drugs on spermatogenesis in cancer patients, aggravating the effects caused by the disease itself [155,156,157,158]. Some commonly used medications, namely psychotropic drugs (imipramine hydrochloride, desmethylimipramine, chlorpromazine, trifluoperazine, and nortriptyline hydrochloride) [159], antiepileptic drugs (phenytoin, carbamazepine, and valproate) [160], acetaminophen (paracetamol) [161], aspirin [162], and lansoprazole [163], have adverse affects on sperm quality. In addition, regular consumption of recreational drugs, such as marijuana, was shown to affect both spermatogenesis and sperm motility [164].

### 3.4. Local Affections of the Genitourinary System

Imbalance of the semen microbiome, varicoceles, orchitis, and prostatitis are the most important local factors that may affect male fertility.

Recent research on the semen microbiome, utilizing sequencing technologies, has consistently shown that semen has its own microbiome, which is far more complex than previously thought, encompassing a diverse range of bacteria with both beneficial and detrimental effects on sperm quality and reproductive outcomes [165,166,167]. Despite its unique nature, semen microbiota show a high interindividual variability resulting from the interplay between environmental factors, personal hygiene habits, and age [168]. There are a number of studies addressing semen microbiota in relation with male fertility and reproductive health. They have been reviewed and critically analyzed recently, and it is shown that, despite much incongruence between data from individual reports, there are also points of agreement [169]. The main bacterial genus/species linked to different pathological conditions include *Neisseria* (semen hyperviscosity and oligoasthenoteratozoospermia) [165], *Klebsiella pneumoniae* (reduced sperm motility and increased apoptosis, semen hyperviscosity [165], *Prevotella* (oligoasthenozoospermia) [170], *Corynebacterium* (asthenoteratozoospermia) [171], *Mycoplasma* and *Ureaplasma* (azoospermia) [172], *Escherichia coli* (sperm acrosome and DNA damage) [173], *Pseudomonas* (oligoasthenoteratozoospermia) [165], and *Ralstonia* and *Stenotrophomonas* (asthenozoospermia) [171]. Interestingly, data obtained by a high-throughput sequencing method targeting V3 and V4 regions of *16S rRNA* gene have shown that, in addition to seminal plasma, the human testis also harbors a potential bacterial signature, though in a low-biomass, and could contribute to the seminal microbiome composition [174].

Varicoceles, orchitis, and prostatitis are other local affections of the genitourinary tract that cause male infertility. A large number of studies have shown that they cause sperm dysfunction, mainly through oxidative stress, and are associated with overproduction of pro-inflammatory cytokines and abnormal spermatogenesis [175]. In addition, a varicocele causes a rise in scrotal temperature with its negative effect on spermatogenesis (see Section 2.4 of this article).

**Table 2 ijms-26-02797-t002:** Selected intrinsic susceptibility conditions and their interactions with external harmful factors. See the main text for a spelling out of the abbreviations used.

Conditions	Examples	Interactions	References
Genetic	*GSTM1* mutation	Decreased air pollutant detoxification	[106,107]
	*XRCC1*, *XRCC1*, *CYP1A1* po1ymorphism	High DNA damage by air pollutants	[107]
	Mutations and deletions of mtDNA	Susceptibility to oxidative stress	[112,113,114]
Epigenetic	*H19* hypomethylation	Susceptibility to effects of smoking	[121,122,123]
Systemic disease and medication	Insulin resistance and diabetes	Potentiation of factors causing oxidative stress and inflammation	[125,126,127,128]
	Infectious diseases	Potentiation of factors causing oxidative stress and inflammation	[136,137,138,139,140,141,142,143,144,145,146,147]
	Chemotherapeutic and antiepileptic drugs, paracetamol, aspirin, lansoprazole, marijuana	Potentiation of factors affecting spermatogenesis and sperm motility	[159,160,161,162,163,164]
Local affections	Semen microbiome	Can exert both potentiating and protective action	[165,166,167,168,169,170,171,172,173,174]
	Varicoceles, orchitis, and prostatitis	Potentiation of factors causing oxidative stress and inflammation	[175]

## 4. Personalized Management

Since the effects of lifestyle and environmental factors on male fertility (see Section 2 of this article) are highly dependent on a particular combination of susceptibility factors (see Section 3 of this article), which is unique to each individual, the management of the resulting pathological condition also needs to be strictly individualized.

### 4.1. Prevention

Among the preventive measures aimed at the reduction in risk for human health damage in general, and reproductive health in particular, collective ones and individual ones can be distinguished. The collective measures are carried out by governments and use habitual tools of centralized regulation, such as promotion of social health-care programs, bans on certain products and industrial procedures, and selective imposition of others. At present, these kinds of measures are mainly focused on the implementation of and increase in health taxes on tobacco, alcohol, and sugar-sweetened beverages, the control of hypertension, medical checkups for the prevention and early detection of cancer, reducing both indoor and ambient air pollution, and, by incentivizing companies and empowering consumers, increasing the consumption of healthy food and decreasing the consumption of unhealthy food [176].

The efficacy of individual preventive measures depends on the willingness and objective capacity of persons exposed to potentially harmful factors to change their habits so as to reduce the load. Dietary habits should be modified to reduce the popular Western-style diet, which is low in vitamins and high in processed products [177]. On the other hand, dietary patterns based on foods rich in antioxidants and anti-inflammatory compounds, such as vitamin C, vitamin E, vitamin D, folate, β-carotene, selenium, zinc, cryptoxanthin, and lycopene, and low in saturated fatty acids and trans-fatty acids appear to help prevent male infertility [177]. Voluntary exposures, such as smoking, alcohol, inadequate exercise, heat, or incorrect positioning of electronic devices (see Section 2 of this article) should be modified. If they cannot be modified easily, such as cases of professional exposures, or if the exposed person is reticent to adopt the lifestyle change recommended, they should be at least taken into consideration. This will make it possible to search for objective markers of their impact in each individual. Since most of the effects of lifestyle and environmental factors converge towards oxidative stress [13,14], essays for the evaluation of oxidative stress markers and their local effects in semen [178,179,180,181] can be considered. A recent study suggested platelet mitochondrial function and endogenous coenzyme Q10 levels as markers of mitochondrial health in infertile men [182]. Data obtained by using such markers can serve as a basis for patient-tailored preventive antioxidant medications and the long-term evaluation of their effects.

### 4.2. Intervention

The consequences of exposure to many lifestyle and environmental factors for male fertility can be reversible. Thus, spontaneous fertility improvement can be observed in many symptomatic men who quit such exposure. For instance, a recent study showed that smoking cessation had a positive effect on sperm concentration, semen volume, and total sperm count [183]. In other cases, treatment of the health issues predisposing people to the effects of external lifestyle and environmental factors, if they can be identified, can also restore fertility or at least improve the existing fertility problem. This was shown to be the case for the use of metformin, a drug employed in both diabetes and insulin resistance without the presence of diabetes, which improved sperm count, morphology, and motility in men suffering from these pathologies [184]. Similarly, beneficial effects of oral antibiotics and anti-inflammatory agents on semen parameters, sperm chromatin structure, and sperm DNA integrity in men with an infertility diagnosis were reported [185], and oral treatment with high doses of antioxidant vitamins C and E improved both sperm DNA integrity [186] and assisted reproduction outcomes [187]. Other medical therapies, including gonadotropins, selective estrogen receptor modulators, and aromatase inhibitors, can also be considered [188]. It is important to note that the population to be treated by these noninvasive measures has to be adequately selected as in inadequately selected patients such therapies might have deleterious effects.

If neither lifestyle modification nor the above noninvasive treatments manage to improve male fertility, the recourse to assisted reproduction (AR) is needed. Depending on the degree of sperm damage, the use of different AR techniques, including intrauterine insemination (IA), conventional in vitro fertilization (IVF), and IVF assisted by intracytoplasmic sperm injection (ICSI) or intracytoplasmic morphologically selected sperm injection (IMSI) can be envisaged. IA and conventional IVF may be used in cases of moderate impairment of sperm count (oligozoospermia) and motility (asthenozoospermia) and of the presence of antisperm isoantibodies in the female partner, while ICSI or IMSI are required for severe oligozoospermia and asthenozoospermia as well as for a high percentage of morphologically abnormal spermatozoa (teratozoospermia). ICSI, developed and first successfully used in 1992 [189], can overcome even the most severe issues of sperm count, motility, and morphology by selecting healthy-appearing spermatozoa and injecting them individually into each oocyte. However, high degrees of sperm DNA fragmentation seriously decrease the chance of livebirth after ICSI by affecting embryo viability and implantation potential without being detectable during the first 3–5 days after fertilization [190]. An empirical study [191] has shown that livebirths in cases of severe sperm DNA fragmentation can be improved by using IMSI, a modification of ICSI introduced in 2001 [192]. Methods for the evaluation of sperm DNA fragmentation are not compatible with sperm survival, so the use of high optical magnifications to select spermatozoa in the context of IMSI cannot detect DNA damage directly [193]. To understand how IMSI can improve AR outcomes as compared to ICSI, it is needed to have a look at the principal DNA-protective mechanism inherent in spermatozoa. In healthy spermatozoa, DNA is shielded against oxidative damage by its close association with protamins in highly condensed sperm chromatin structures. This protection is ineffective in regions in which sperm chromatin condensation is defective, and this condition is reflected by the presence of intranuclear vacuoles that can be observed by electron microscopy (Figure 2A). Even large intranuclear vacuoles are mostly undetectable with optical magnifications used in conventional ICSI but they do appear at the magnification used in IMSI (Figure 2B) so that injection of spermatozoa with such vacuoles into oocytes can be avoided [191,193]. If the nocive action of a given lifestyle or environmental factor leads to an irreversible blockage of germ cell development at the round spermatid stage, round spermatid injection (ROSI) into oocytes may represent the last-chance treatment [194]. Pregnancies and births were achieved with the use of both freshly recovered spermatids [194,195] and those resulting from in vitro development of primary spermatocytes into spermatids [196]. Besides IVF, ICSI, IMSI, and ROSI, varicocelectomy is another invasive intervention to be considered; it was reported to improve semen quality and fertility in men with a varicocele [175], though there is no common agreement on this effect.

## 5. Conclusions

Under contemporary lifestyle and environmental conditions, male fertility can be affected by a number of potentially harmful factors related to unhealthy habits or involuntary professional or nonprofessional exposures. The degree of the resulting fertility impairment is highly individual and is given by an interplay between the external factors and an inherent susceptibility to them. If a man is exposed to a known external factor, his susceptibility should be evaluated by appropriate diagnostic methods. In many cases, the damage caused to the reproductive system is reversible and can be repaired spontaneously when the action of the factor(s) in question ceases. If the exposure cessation is difficult to achieve (obesity, professional exposures, stress) or accept (smoking, alcohol), or if the causative agent comes from sources independent of individual human will (air pollution, harmful chemicals present in food, beverages, cosmetics, etc.), symptomatic treatment, mainly based on individualized oral intake of antioxidants, can be envisaged. In the most severe cases, the recourse to assisted reproduction techniques is possible.

## Figures and Tables

**Figure 1 ijms-26-02797-f001:**
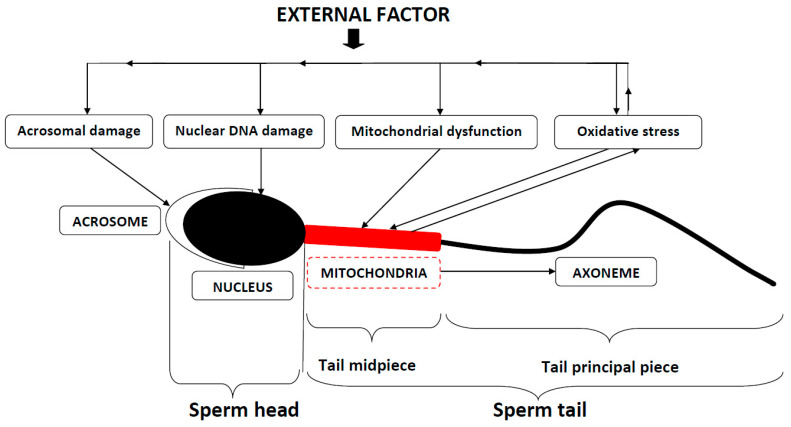
Schematic representation of a human spermatozoon showing how it can be affected by the action of external factors.

**Figure 2 ijms-26-02797-f002:**
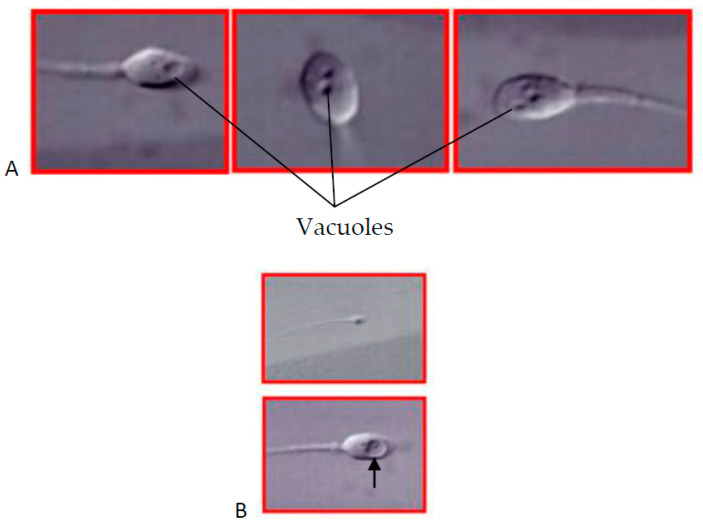
(**A**) Scanning electron micrographs of human spermatozoa with different-sized intranuclear vacuoles. (**B**) A large intranuclear vacuole observed in a living human spermatozoon at magnifications used in ICSI (upper image, barely seen) and IMSI (lower image, arrow) settings. Adapted from Tesarik [116].

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
