# Peer review of "Lifestyle and Environmental Factors Affecting Male Fertility, Individual Predisposition, Prevention, and Intervention"

_ijms, 2025, doi:10.3390/ijms26062797_

Round 1

Reviewer 1 Report

Comments and Suggestions for Authors

Dear Author

I read your review ‘Lifestyle and Environmental Factors Affecting Male Fertility, Individual Predisposition, Prevention and Intervention’. The topic is certainly topical and of interest, I only have a few comments to make:

- In the section on obesity a large part of the speech is repeated, this is a typo that needs to be corrected.

- It seems to me excessive to state with certainty that global warming damages spermatogenesis, it could be said that this could be evaluated with interest in future studies.

- In the section on stress I would better define the concept: stress resulting from what conditions? Physical stress? Psychological stress?

- In the section on interventions when talking about antioxidants and antibiotics I would emphasise more the importance of adequately selecting the population to be treated as in inadequately selected patients such therapies could have deleterious effects.

- Before discussing IVF techniques, I would mention the role of medical therapies with FSH, SERMs and aromatase inhibitors.

- At least in the section related to the impact of electromagnetic fields I suggest citing the following paper https://doi.org/10.1016/j.tice.2023.102045 that is more recent than the references listed in the review.

Best regards

Author Response

Dear Reviewer

Thank you very much for your comments which will certainly increase the manuscript quality. I have included the corresponding modifications in the manuscript (highligted in red). This is a list of the changes made.

- In the section on obesity a large part of the speech is repeated, this is a typo that needs to be corrected.

Response: The repeated part of this section has been removed.

- It seems to me excessive to state with certainty that global warming damages spermatogenesis, it could be said that this could be evaluated with interest in future studies.

Response: The text has been changed, and a sentence has been added  to say that  “the question of whether global warming also can play a role is to be evaluated with interest in future studies”.

- In the section on stress I would better define the concept: stress resulting from what conditions? Physical stress? Psychological stress?

Response: A sentence stating that stress can result from both physical and psychological conditions has been added. As to the physical stress, the reader is referred to the section 2.8. “Inadequate Physical Activity”.

- In the section on interventions when talking about antioxidants and antibiotics I would emphasise more the importance of adequately selecting the population to be treated as in inadequately selected patients such therapies could have deleterious effects.

Response: A sentence has been added stating that the population to be treated by these noninvasive measures has to be adequately selected as in inadequately selected patients such therapies might  have deleterious effects.

- Before discussing IVF techniques, I would mention the role of medical therapies with FSH, SERMs and aromatase inhibitors.

Response: A sentence has been added  to note that other medical therapies, including gonadotropins, SERMs and aromatase inhibitors, can also be considered.

- At least in the section related to the impact of electromagnetic fields I suggest citing the following paper https://doi.org/10.1016/j.tice.2023.102045 that is more recent than the references listed in the review.

Response: The suggested citation has been added.

Best regards

Jan Tesarik

Reviewer 2 Report

Comments and Suggestions for Authors

The manuscript presents a comprehensive and well-structured review of how various lifestyle and environmental factors contribute to male fertility impairment. It thoroughly discusses key contributors such as obesity, smoking, alcohol consumption, air pollution, heat exposure, and chemical contaminants, while also considering genetic and epigenetic predisposition. The topic is timely and relevant, addressing the growing concerns about declining male fertility. The review integrates multiple factors into a holistic discussion and is supported by numerous citations that add credibility to the claims made. The manuscript is logically structured, moving from general lifestyle factors to more specific environmental and genetic considerations. Additionally, the inclusion of figures and tables enhances the understanding of key mechanisms affecting sperm function and fertility.

However, several areas require improvement to enhance clarity, scientific depth, and engagement with the latest research:

  1. clarity and focus: Some sections, particularly those discussing epigenetic influences and genetic susceptibility, require clearer explanations and better-defined subheadings to improve readability. The discussion on oxidative stress as a unifying mechanism is useful but would benefit from a concise summary or a visual representation.

  2. scientific depth and recent literature: While the manuscript is well-referenced, it lacks some recent studies from the past two to three years. Updating references to include recent findings on sperm epigenetics, oxidative stress, and intervention strategies would strengthen the discussion. The section on mobile phones and electromagnetic radiation effects on sperm quality should also consider more recent meta-analyses and systematic reviews.

  3. methodological considerations: Although this is a review paper, it would be helpful for the author to briefly discuss the criteria for including or excluding studies. Were meta-analyses preferred over individual studies? How were conflicting results reconciled? Additionally, when discussing environmental exposures, including specific data on exposure levels that are considered harmful would add clarity.

  4. figures and tables: Some figures lack clear labels or detailed explanations in the figure legends. Improving figure clarity and ensuring they align with the corresponding text discussion would enhance comprehension.

The manuscript is well-prepared but requires minor revisions to improve clarity, update references, and refine figures. After these adjustments, it will be suitable for publication in International Journal of Molecular Sciences.

Comments on the Quality of English Language

While the manuscript is generally well-written, some sentences are overly complex and would benefit from simplification. Minor grammatical and typographical errors are present; a thorough proofreading or professional editing service is recommended.

Author Response

Dear Reviewer

Thank you very much for your comments which will certainly increase the manuscript quality. I have included the corresponding modifications in the manuscript (highligted in red). This is a list of the responses and changes made.

  1. The sections dealing with genetic and epigenetic factors have been separated in two distinct subheadings, and new citations of recent research have been added.
  2. References to recent findings on sperm epigenetics, oxidative stress, intervention strategies, mobile phones and electromagnetic radiation effects have been added..
  3. A section “Methodological Considerations” has been added after “Conclusions”. Both meta-analyses and individual studies were included, based exclusively on their impact and novelty. Eventually found conflicting results were included and discussed . As to specific data on harmful exposue levels for individual lifestyle and environmental factors, they have ben included where available (eg., heat exposure). However, most of the factors studied are not measurable or their threshold levels considered to be harmful for male fertility are subjective in the context of environmental exposure (e.g., obesity, stress, alcohol, physical activity, electromagnetic fields).
  4. Figure 2 and the corresponding legend have been modified to make clear what is meant by “intranuclear vacuoles”.
  5. The whole text has been proofread, overly complex sentences have been split, and grammatical and typographical erros have been corrected.

Best regards

Jan Tesarik